# Theoretical and Numerical Study on the Pile Barrier in Attenuating Seismic Surface Waves

**Chunfeng Zhao [1,2,*], Chao Zeng [2], Yinzhi Wang [2], Wen Bai [1] and Junwu Dai [1]**

[1] Key Laboratory of Earthquake Engineering and Engineering Vibration, Institute of Engineering Mechanics, China Earthquake Administration, Harbin 150080, China
[2] College of Civil Engineering, Hefei University of Technology, Hefei 230009, China
\* Correspondence: zhaowindy@hfut.edu.cn

**Abstract:** The purpose of this study is to investigate the attenuation effect of the pile barrier in blocking seismic surface waves by using theoretical and numerical methods. First, we derive the dispersion characteristics of pile barriers embedded in soil from the perspective of periodicity theory to explain that such periodic barriers can attenuate seismic surface waves when the main frequencies fall into the band gaps of the pile barrier. Second, the dispersion characteristics of periodic barriers composed of different inclusions are discussed, and it is suggested preliminarily that scatters with low stiffness and low density are more conductive to mitigate low-frequency surface waves. Third, a three-dimensional transmission calculation model is also developed to illustrate that the attenuation zone of a finite number of piles is consistent with the surface wave band gap. Finally, transient analysis of the periodic pile barriers is performed to validate the block effects on seismic surface waves. The numerical results show that the frequency band gaps of multi-row pile barriers are in accordance with the frequency band gaps of the surface wave in theory, which can greatly mitigate surface ground vibration. The pile spacing, number of piles, and pile length are the key parameters that can affect the width of attenuation zones of the periodic barriers by an appropriate design.

**Keywords:** frequency band gap; isolation effect; seismic surface waves; pile barrier; transient analysis

## 1. Introduction

The vibrations propagating in the underground and at the ground surface are generally classified into body waves and surface waves [1]. A large number of measured seismic records indicate that the predominant frequencies of surface waves range from 1 to 20 Hz. Generally, surface waves carry most of the vibration energy, with exponentially decaying amplitudes along the free surface in an elastic semi-infinite space, and travel slower than body waves, which may cause more damage to surrounding buildings [2–7]. Simultaneously, surface energy reflection and redirection may further lead to a more intense impact than bulk waves (BWs). Thus, it is necessary to develop new techniques to mitigate the ground vibration caused by surface waves, mainly Raleigh waves.

In fact, various wave barriers, including open and infilled trenches [8], rows of holes [9], and pile barriers [10], have been experimentally and numerically investigated in manipulating and attenuating the propagation of elastic waves. The results show that such passive isolation devices, periodically constructed between the source of the vibration and the protected buildings, are expected to be an effective technique to reduce the undesired ground motions and prevent considerably the structure from the threat of strong vibrations. Based on a series of experiments, Wood et al. [11] proposed the basic principle of using rows of piles for seismic isolation design, and assessed the isolation performance of the wave barriers. In the following decades, many researchers conducted numerous experiments to study the shielding effect of wave barriers. Celebi et al. [12] carried out field tests on foundation vibrations to evaluate the isolation effectiveness of various types

of wave barriers including open and infilled trench barriers. Naggar et al. [13] used an experimental method to examine the attenuation effectiveness of open and geofoam-filled trenches and discussed the influence of exciting frequency and source distance on the attenuation effects. Additionally, Huang et al. [14] applied the large-scale field test to evaluate the isolation effect of both empty trenches and periodically arranged continuous walls, and the results showed that three types of excitation inputs yielded similar results, namely, the performance of periodic barriers depends on the attenuation frequency range generated by different excitation directions.

After that, numerical methods were used to investigate the protective effectiveness of wave barriers. For instance, Saikia et al. [15] numerically presented a two-dimensional (2D) plane strain model to analyze the blocking efficiency of open trenches on the surface of linear elastic soil media subjected to vertical harmonic excitation. The numerical results showed that the shielding effect was determined by the normalized depth of the open trenches and independent of width in most cases. Celebi and Kırtel [12,16] investigated the blocking performance of trench-type barriers in reducing the ground vibration induced by trains. Subsequently, Zoccali et al. [8] presented a 3D FE model to evaluate the screening effectiveness of trenches vibrations caused by railway traffic both in time and frequency domain and found that the attenuation efficiency is closely related to the trench length and infilled material types.

Some researchers have proposed the possibility of using various new periodic structures to shield bulk waves and surface waves in civil engineering. Zhao and Witarto et al. [17–20] constructed a new kind of 1D layered periodic foundation using rubber and concrete, and studied experimentally and numerically the isolation effectiveness of the periodic foundation on blocking longitudinal and shear waves in the frequency range of 0–50 Hz. Brule et al. creatively proposed a real-size experiment with periodic cylindrical bore holes at meter-size to block the seismic surface waves around 50–100 Hz [9]. Huang et al. [21] verified that the simulated vibration reduction frequency segments are consistent with the theoretical surface-wave attenuation zones through a 3D FE transmission model. Pu and Shi also studied the surface wave isolation of a pile barrier, and the effects of soil property, pile spacing, and pile length on the attenuation zone of the wave barriers were also considered [10]. Moreover, Palermo et al. proposed a method where resonant barriers buried under the soil surface could manipulate the band gap below 10 Hz and isolate the seismic surface waves effectively [22].

Although some works have focused on the isolation effectiveness of the wave barrier in blocking surface waves in the past, the isolation performance of the periodic pile barriers deserves further investigation. First, little work has been focused on the isolation performance of multi-row pile barriers based on the dispersion properties of Rayleigh wave propagation in periodic piles. Second, the time history analyses of pile barriers have been rarely conducted to verify the attenuation characteristics. Therefore, this work aims to investigate surface wave mitigation by using multi-rows of pile barriers.

The rest of the work is arranged as below. The dispersion equation of an elastic wave is derived using the wave governing equation and Floquet–Bloch boundary conditions in Section 2. The attenuation characteristics of surface waves are discussed in detail, including the identification method of Rayleigh modes and the isolation performance of a finite array of piles in Section 3. In Section 4, the effects of pile geometry, number of piles, and focal distance on the surface wave band gap are studied parametrically. In Section 5, transient analyses of an artificial wave and seismic wave are adopted to verify the attention performance of the periodic pile barriers for surface wave mitigation. Last of all, in Section 6, the conclusions are provided.

## 2. Theoretical Framework for Periodic Structures

*2.1. Wave Equation*

For wave propagation in an isotropic, homogeneous linear elastic medium, the vibration modes of the material conform to the following governing equation:

$$\rho \frac{\partial^2 \mathbf{u}}{\partial t^2} = \nabla[(\lambda + 2\mu)(\nabla \cdot \mathbf{u})] - \nabla \times [\mu \nabla \times \mathbf{u}] \tag{1}$$

where $\mathbf{u} = \mathbf{u}(\mathbf{r})$ is the displacement vector, $\rho$ represents the material density, $\lambda$ and $\mu$ are Lame parameter functions of the medium, which can be expressed by Young's modulus $E$ and Poisson's ratio $v$, respectively, as follows:

$$\lambda = \frac{vE}{(1+v)(1-2v)} \tag{2}$$

$$\mu = \frac{E}{2(1+v)} \tag{3}$$

It is well known that Equation (1) can be decoupled into two sets of mutually independent equations describing out-of-plane (Equation (4)) and in-plane fluctuations (Equations (5) and (6)), when the elastic wave propagates only in the *xOy* plane. Here, we mainly consider surfaces waves propagating along with elastic semi-infinite space, so the governing equation should be based on Equations (5) and (6).

$$\rho \frac{\partial^2 u_z}{\partial t^2} = \frac{\partial}{\partial x}(\mu \frac{\partial u_z}{\partial x}) + \frac{\partial}{\partial y}(\mu \frac{\partial u_z}{\partial y}) \tag{4}$$

$$\rho \frac{\partial^2 u_x}{\partial t^2} = \frac{\partial}{\partial x}\left[\lambda(\frac{\partial u_x}{\partial x} + \frac{\partial u_y}{\partial y})\right] + \frac{\partial}{\partial x}\left[\mu(\frac{\partial u_x}{\partial x} + \frac{\partial u_x}{\partial x})\right] + \frac{\partial}{\partial y}\left[\mu(\frac{\partial u_x}{\partial y} + \frac{\partial u_y}{\partial x})\right] \tag{5}$$

$$\rho \frac{\partial^2 u_y}{\partial t^2} = \frac{\partial}{\partial y}\left[\lambda(\frac{\partial u_x}{\partial x} + \frac{\partial u_y}{\partial y})\right] + \frac{\partial}{\partial x}\left[\mu(\frac{\partial u_y}{\partial x} + \frac{\partial u_x}{\partial y})\right] + \frac{\partial}{\partial y}\left[\mu(\frac{\partial u_y}{\partial y} + \frac{\partial u_y}{\partial y})\right] \tag{6}$$

*2.2. Periodic Boundary Conditions and Eigen Equation*

Figure 1a shows the schematic representation of the pile barriers embedded periodically in elastic half-space. The multi-row of embedded piles can be assumed as a two-dimensional periodic pile configuration. For a two-dimensional (2D) periodic pile-soil configuration, the dispersion relations can be calculated by a unit cell, periodic boundary conditions (PBCs) are used as the lateral boundaries, as depicted in Figure 1b. To minimize computation, it is assumed that piles and soil are homogeneous, isotropic, and perfectly bonded at the interface. As shown in Figure 1c, the pile radius, pile length, and soil thickness can be expressed as r, $h_1$, and $h_0$, respectively. The displacement vector $\mathbf{u}$ in Equation (1) can be expressed as

$$\mathbf{u}(\mathbf{r}, t) = e^{\mathrm{i}(\mathbf{k} \cdot \mathbf{r} - \omega t)} \mathbf{u_k}(\mathbf{r}) \tag{7}$$

where, $\mathbf{r}$ is the position vector, $i = \sqrt{-1}$, $\omega$ and $\mathbf{u_k}(\mathbf{r})$ represent the angular frequency and a modulation function of the displacement vector, $\mathbf{k}$ is the Bloch wave vector. The modulation function can be written as:

$$\mathbf{u_k}(\mathbf{r}) = \mathbf{u_k}(\mathbf{r} + \mathbf{R}) \tag{8}$$

where $\mathbf{R}$ represents the lattice constant vector, $\mathbf{R} = (\mathbf{Rx}, \mathbf{Ry})$. For the 2D periodic piles arranged in the quadrate configuration, $\mathbf{Rx} = \mathbf{Ry} = \mathbf{R}$.

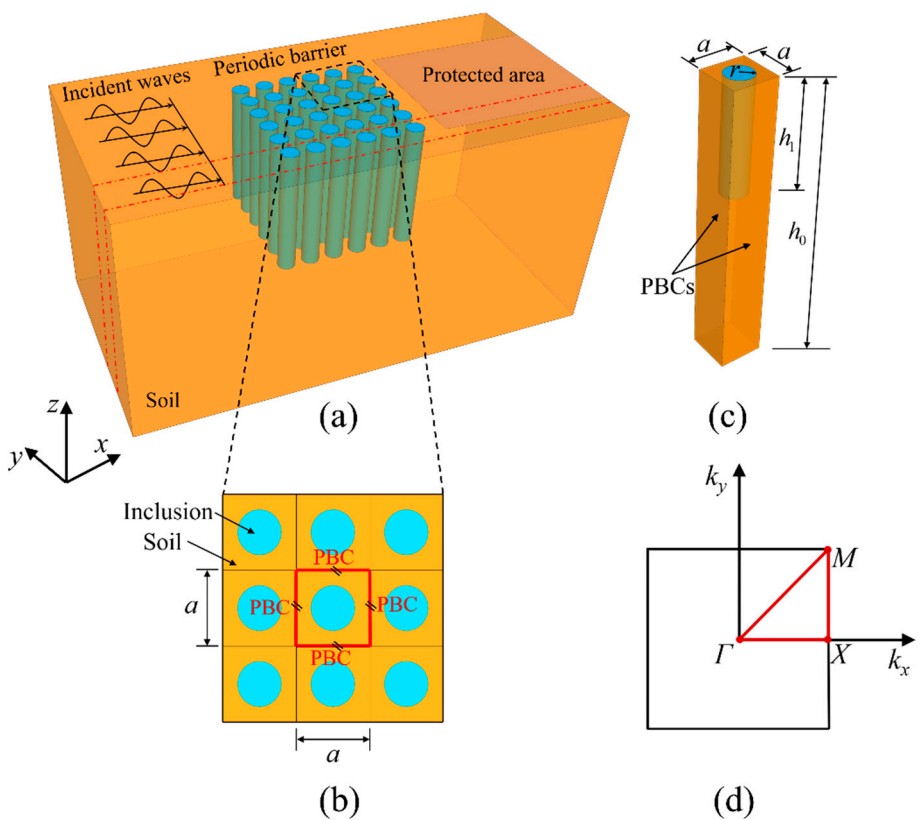

**Figure 1.** Schematic diagram of the proposed periodic pile barriers. (**a**) Rows of the piles embedded in an elastic half-space; (**b**) plan view; (**c**) typical unit cell; (**d**) the first irreducible Brillouin zone and corresponding irreducible Brillouin zone.

Submitting Equation (8) into Equation (7), the PBCs can be represented in the form

$$\mathbf{u}(\mathbf{r} + \mathbf{R}, t) = e^{i\mathbf{k} \cdot \mathbf{R}} \mathbf{u}(\mathbf{r}, t) \tag{9}$$

Additionally, it should be noted that the motion of Rayleigh waves decays rapidly with depth and the displacement and surface wave energy are mainly concentrated within the 1.5 wavelength range. Based on these features, soil with finite depth and clamped bottom boundary conditions is considered to simulate semi-infinite space. After referring to the previous works [10,23], the unit cell with a height $h_0 = 20a$ is adopted in this paper, which ensures capture of the Rayleigh modes around the resonance frequency, a is the length of unit cell. The validity of this method is verified in Section 3.

By substituting the PBCs of Equation (9) into Equation (1), we can obtain the dispersion relation of an infinite periodic structure and the eigenvalue equation can be expressed as follows

$$\left( \mathbf{K} - \omega^2 \mathbf{M} \right) \mathbf{u} = 0 \tag{10}$$

In which **K** and **M** represent the stiffness and mass matrices of the unit cell, respectively. It is worth noting that the eigenvalue equation can be considered mathematically as an implicit function between the wave vector k and angular frequency $\omega$, where the corresponding frequency zones of the wave vector do not exist are frequency band gaps.

COMSOL Multiphysics software is used to calculate the dispersion relation of the periodic structure. A single pile embedded in portion soil is assumed as a typical unit cell, which can be simulated by 4-node tetrahedral elements. A fixed boundary is applied at the bottom surface of the unit cell and the Floquet PBCs in Equation (9) are adopted at its vertical sides. Consequently, all the propagating modes can be obtained along the Γ- X- M-

Γ in the irreducible Brillouin zone (see in Figure 1d) and thus the dispersion curves can be plotted.

## 3. Attenuation Properties of Surface Waves

This section is divided by subheadings. It should provide a concise and precise description of the experimental results, their interpretation, as well as the experimental conclusions that can be drawn.

### 3.1. The Identification Method of Rayleigh Modes

It should be noted that the dispersion relations calculated by COMSOL above include both surface modes and bulk modes polarized along all directions. To identify the Rayleigh wave modes from the bulk wave modes, we used the post-processing method proposed by Huang [21] to calculate the surface wave attenuation zone of the layered periodic structures. As described in their previous works, the modes of the surface wave can be easily distinguished from all the mixed eigenmodes because the elastic strain energy of Rayleigh is mainly concentrated near the surface. After identifying all surface wave modes within the frequency range of interest, the dispersion laws of Rayleigh waves can be obtained by plotting the corresponding relation between wave vector and frequency.

Here, an energy distribution parameter $\xi$ is defined based on the strain energy center with the depth of the unit cell, indicating the energy distribution of a considered eigenmode along the depth. In particular, the energy distribution function of the unit cell can be written as follows:

$$\xi = \frac{\int_0^{h_0} z \cdot W_s dV}{h_0 \int_0^{h_0} W_s dV} \tag{11}$$

where the integral region is limited to the unit cell volume $V$, $h_0$ represents the unit cell height, and $W_s$ represents the elastic strain energy density. It can be found from Equation (11) that the distribution parameter varies from 0 to 1. What is more, a larger magnitude of distribution parameter indicates that the corresponding mode energy localizes near the free surface. In this paper, Rayleigh wave modes are assumed on the condition of $\xi \geq 0.9$, while the other modes with a distribution parameter less than 0.9 are eliminated.

### 3.2. Comparision and Verification

The accuracy of the aforementioned criterion for identifying surface waves is verified by comparing the numerical results with the previous studies [24,25]. Figure 2 shows the dispersion relations of surface waves and bulk waves for the layered Cu-Al periodic structure. The dispersion curves of Longitudinal waves (P waves) and Rayleigh waves (R waves) are calculated by the transfer matrix method and a post-processing technique using COMSOL respectively. The P waves are represented by black dotted lines and R waves by solid brown lines. It is observed that the dispersion curves are in accordance with the results calculated by previous works, which indicates that the present method has good accuracy in calculating the dispersion curves for surface waves and bulk waves.

The right panel of Figure 2 demonstrates the vibration modes of a typical unit cell obtained from the correlation dispersion curves. Obviously, it can be found that the amplitudes of the displacement for the two surface modes are concentrated on the near free surface. Additionally, it can be seen that the normalized displacement field of modes A and B are of sin-like symmetry, which is a typical characteristic of Rayleigh waves. The first two surface wave band gaps (SWBGs) are also plotted in Figure 2, showing that the designed layered periodic structure can be expected to block the surface waves within the band gaps.

Taking another example published in related research [24], a 3D numerical model can be established to investigate the dispersion relations of a surface wave (SW) in a foam pile-soil system. In the model, the parameters of density, Young's modulus, and Poisson ratio of soil are 1800 kg/m$^3$, 46 MPa, and 0.25, and the corresponding parameters for foam

are 60 kg/m$^3$, 37 MPa, and 0.32, respectively. The values of a = 0.8 m and r = 0.3 m represent the periodic constant and the radius of the pile. Figure 3 shows the dispersion curves of SWs in the periodic structure and its transmission loss spectrum. The results from the present model are denoted by a black solid line and the reference results are represented by pink solid circles. As expected, the results are highly consistent, which again proves the reliability of the surface wave recognition criterion.

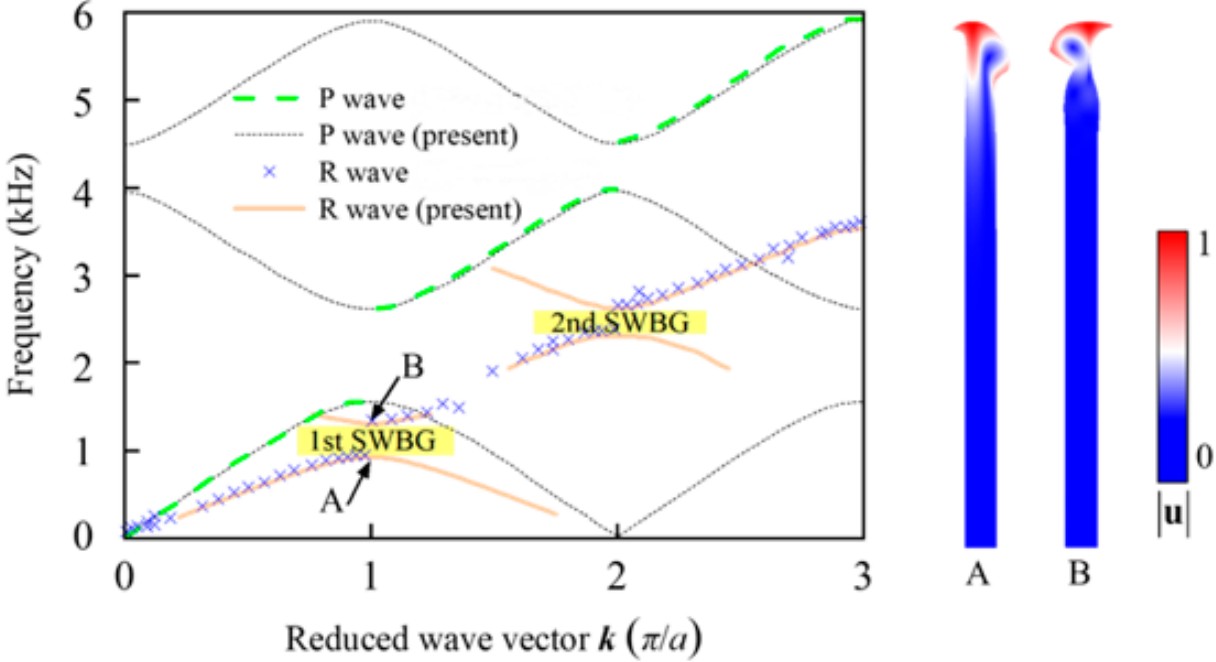

**Figure 2.** Dispersion relation between surface waves and bulk waves for the layered Cu-Al pe-riodic structure and corresponding vibration modes, green point and blue cross symbol represent the result in [21,25].

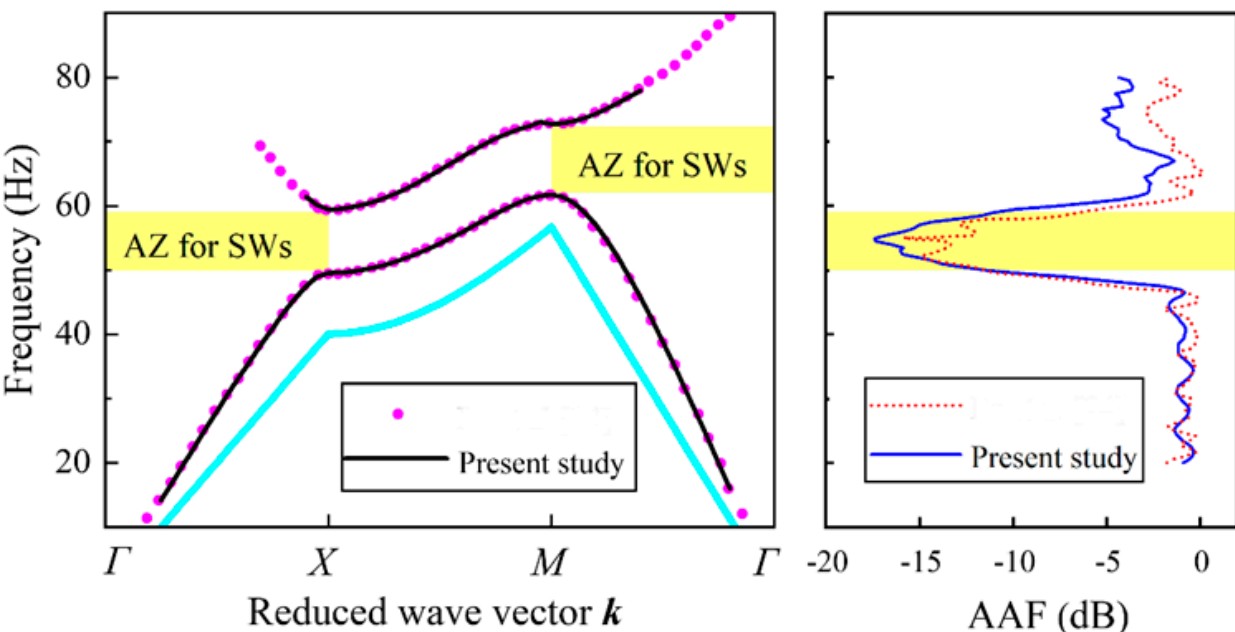

**Figure 3.** Dispersion relation of SWs in the periodic pile−soil system and its transmission loss spectrum., the pink points represent the result in [24].

Considering the unique properties of SW propagation, some scholars proposed distinguishing SWs by using the acoustic cone criterion [26,27]. The sound lines are a function of the wave vector and phase velocity obtained by scanning the wave vector in the first Brillouin zone. Therefore, the SW modes only occur in the dispersion curves below the acoustic cone of the substrate, which can be used as the basis for identifying the SW bandgaps (SWBGs).

To this aim, Figure 3 also superimposes the acoustic cone as a continuous cyan line. The results show that all SW modes occur above the sound line, which means that the sound cone criterion cannot recognize the SW modes with piles embedded under the soil surface. Additionally, as shown in the transmission analysis on the right of Figure 3, the frequency zone in which the output surface displacement amplitude decreases significantly is consistent with the SWBG in the Γ-X direction. The FE model for transmission analysis of finite unit cells is provided in the following paragraphs. In summary, the above two cases fully demonstrate that the method based on strain energy distribution parameters can accurately identify all Rayleigh modes, which is more reliable for calculating the dispersion curve of surface waves of scattered periodic structures.

### 3.3. Dispersion Analysis for Surface Waves in Periodic Pilers

It has been shown that the mechanical properties and geometry dimensions have a strong influence on the attenuation zones of Bragg scattering periodic structures [28,29]. Only a reasonable periodic structure design can obtain low initial frequency and broadband attenuation. In previous research, Huang [21] and Achaoui et al. [30] investigated the shielding effect of 1D periodic trench barriers and 2D piles with different inclusions on surface waves, respectively. Consequently, these inclusion materials are also applied to simulate the periodic piles in this paper, as shown in Table 1.

**Table 1.** Mechanical properties of piles and soil.

| Material | Young's Modulus (MPa) | Density (kg/m$^3$) | Poisson Ratio |
|---|---|---|---|
| Soil | 46 | 1800 | 0.25 |
| Polyfoam | 11.8 | 80 | 0.4 |
| EPS geofoam | 37 | 60 | 0.32 |
| Concrete | 11,316 | 2400 | 0.25 |
| Fly ash | 25 | 500 | 0.35 |
| Steel | 200,000 | 7850 | 0.33 |

To study the influence of different inclusions on the initial frequency and bandwidth of the first SWBG, the low and high bound frequencies of the first SWBG are defined as the lower bound frequency (LBF) and upper bound frequency (UBF), respectively. The width of the SWBG is represented by WBG, which is equal to the value of the UBF minus LBF. The physical model in Figure 1c is reconsidered in this section.

Taking the lattice constant, a = 3 m, pile radius r = 1.2 m, pile length $h_1$ = 2a, and soil column height $h_0$ = 20a as an example, Table 2 shows the first SWBG of periodic piles with five different configurations. It should be noted that the band gap characteristics obtained in Table 2 are only the results of wave vector scanning along the Γ-X direction, which is the focus of this paper. When the incident surface waves propagate in a periodically arranged pile barrier, destructive interference leads to the SWBGs, which greatly attenuates the wave amplitude. As pointed out by Huang et al. [28], the frequency components of SWBGs are mainly dominated by the stiffness and density of the inclusion materials. It is indicated in Table 2 that the different inclusion materials can produce various SWBGs and WBGs; the inclusion material polyfoam can generate the smallest LBF (12.91 Hz) while the inclusion material of concrete in pile barriers has the widest WBG (17.78 Hz). Considering the low initial frequency and broadband attenuation, polyfoam of the same radius is more suitable as a periodic barrier to block seismic surface waves. Therefore, the polyfoam is applied

as inclusion material in the following dispersion analysis of unit cell and transmission analysis of finite rows of pile respectively.

**Table 2.** The first surface wave band gaps of different inclusions.

| Inclusion | LBF (Hz) | UBF (Hz) | WBG (Hz) |
|---|---|---|---|
| EPS geofoam | 17.41 | 22.76 | 5.62 |
| Polyfoam | 12.91 | 18.39 | 5.48 |
| Concrete | 31.23 | 49.01 | 17.78 |
| Fly ash | 14.68 | 18.54 | 3.86 |
| Steel | 36.24 | 49.16 | 12.82 |

Figure 4 illustrates the first SWBG and dispersion relation of Rayleigh waves in polyfoam pile barriers. The yellow shaded areas correspond to the frequency range of SWBG. It is found that two directional SWBGs appear in the direction of ΓX and MΓ, which is different from the all-directional SWBG of periodic piles arranged on the soil surface [31]. Analogously, the four eigenmodes denoted as points A, B, C, and D are presented in Figure 5. It indicates that the normalized displacement fields corresponding to all these eigenmodes are focused on a significant thin region near the free surface. The vibration modes A and B also exhibit sin-like and cos-like characteristics similar to the Rayleigh mode of layered periodic structure mentioned above. The modes C and D also show similar results.

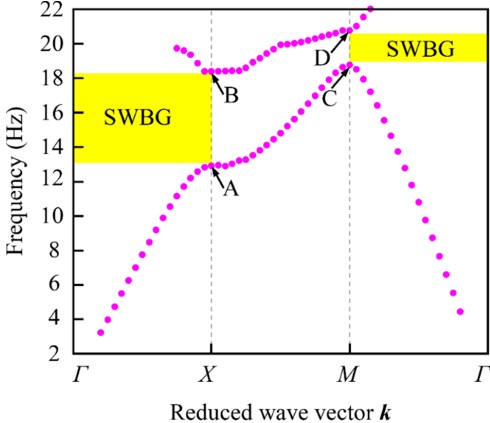

**Figure 4.** Frequency band gaps for Rayleigh waves of periodic polyfoam pile barriers embedded in homogeneous soil with $a = 3$ m, $r = 1.2$ m, $h_0 = 15a$, and $h_1 = 2a$.

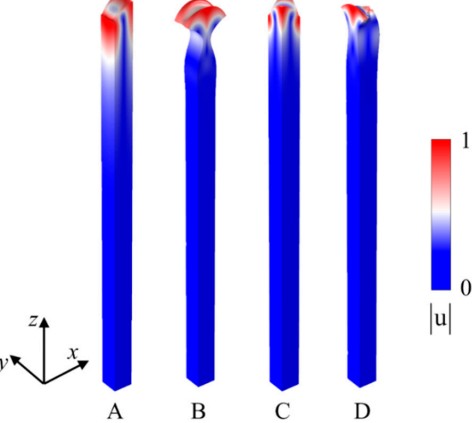

**Figure 5.** The mode shape of points A, B, C, and D in Figure 4.

To get further insight on the decaying characteristics of SWs, Figure 6 shows the distribution of normalized elastic strain energy and displacement amplitude of surface modes along the depth. It is observed that all the energy and displacement of surface modes at points A, B, C and D rapidly decay to zero when the pile barrier depth ratio ($z/h$) approaches 0.2, which indicates that the identified modes are indeed SW modes.

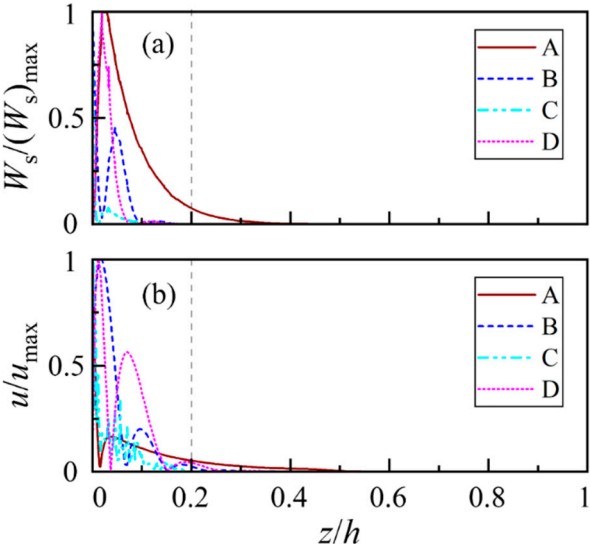

**Figure 6.** The energy and displacement distribution of mode shapes along the depth. (**a**) The normalized elastic strain; (**b**) the normalized amplitude of the displacement.

### 3.4. Vibration Mitigation Efficiency of Finite Unit Cells

In theory, a periodic structure with infinite unit cells possesses a perfect attenuation performance for the vibrations. In practical engineering, the periodic structure consists of finite unit cells. To demonstrate the effectiveness of the finite number of unit cells on mitigation of the SWs, a 3D transmission model with six rows of piles is used to verify the vibration mitigation effectiveness of surface ground vibration. Since the pile barriers are also periodically arranged along the y-direction, a pair of PBCs can be applied perpendicular to the y-axis to save calculation cost. The front view and top view of the 3D FE model is shown in Figure 7. All the geometrical features of the numerical model are $l_0 = 27a$, $l_1 = 10a$, $l_2 = 3a$, $h_0 = 20a$, and $h_1 = 2a$. The pile line spacing is equal to the periodic constant a, the output zone with an area of a is located behind the last pile barrier with a distance of $l_2$.

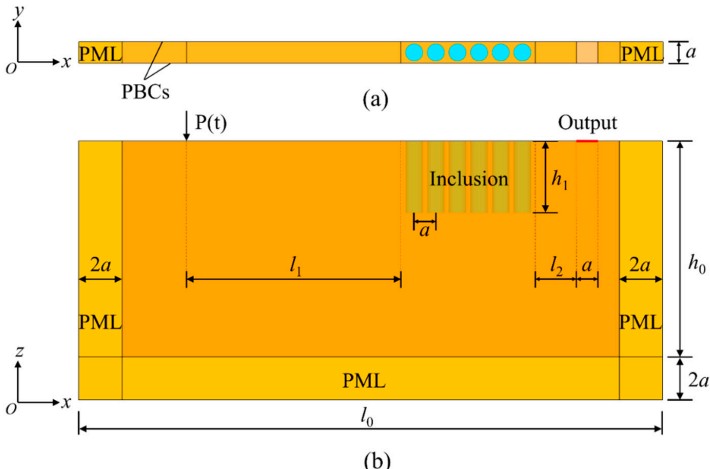

**Figure 7.** Numerical model of multi-rows of pile barriers. (**a**) Front view; (**b**) vertical view. ($l_0 = 27a$, $l_1 = 10a$, $l_2 = 3a$, $h_0 = 20a$, and $h_1 = 2a$).

In the numerical model, a vertical harmonic line source P(t) is triggered with a distance $l_1$ away from the first line of piles which may generate both BWs traveling in the soil substrate and SWs propagating along the surface. In the region far from the source, the surface displacement is mainly caused by SWs and the influence of the BWs is negligible. Perfect matched layers (PMLs) with thickness 2$a$ are applied to the two ends and bottom of the numerical model, which can eliminate the effect of the reflected waves [32–34]. Additionally, an identical transmission model without pile barriers is adopted to obtain the reference displacement field.

An average amplitude reduction factor $\overline{A}_R$ is defined to assess the attenuation effectiveness of the pile barriers.

$$\overline{A}_R = \frac{\int_0^S u_0 dS}{\int_0^S u_1 dS} \tag{12}$$

where $u_0$ and $u_1$ represent the vertical displacement with and without barriers in the output area, respectively. Further, the amplitude attenuation function (AAF) can be written as follows:

$$\text{AAF} = 20 \log_{10}(\overline{A}_R) \tag{13}$$

In fact, the AAF has an explicit mathematical meaning. Its magnitude represents the attenuation capacity of the pile-soil barrier system. Taking AAF = −20 as an example, it indicates that the displacement amplitude of the periodic barrier system in the output area is about 1/10 of that without the periodic barrier system.

Figure 8 shows the transmission attenuation curves as well as the dispersion curves for six rows of polyfoam pile barriers. As shown in Figure 8, the frequency region in which the surface displacement amplitude decreases significantly (AAF ≤ −10), is consistent with the SWBG of the ideal periodic pile barrier. Simultaneously, the transmission spectrum shows that six rows of polyfoam pile barriers generate an expected significant reduction of SWs in its SWBG.

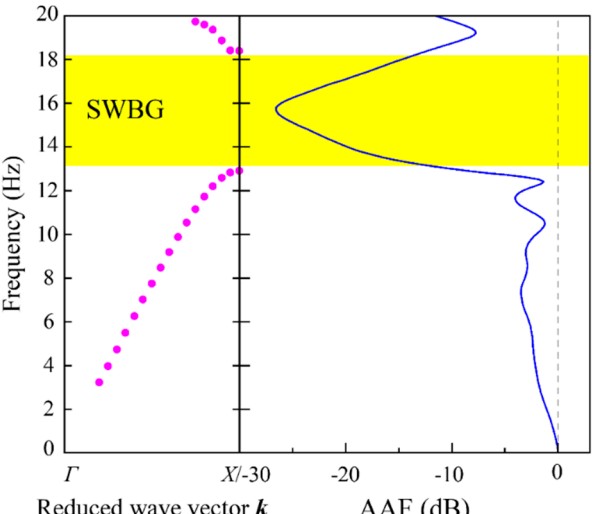

**Figure 8.** The amplitude attenuation function for six rows of polyfoam pile barriers and dispersion curves. ($a$ = 3 m, $r$ = 1.2 m, $l_1$ = 10a, $h_0$ = 15a, and $h_1$ = 2a).

To highlight the generation mechanisms of SWBGs, Figure 9 depicts the normalized vertical displacement response $u_z$ at the free surface and the color-coded wave field distribution of six rows of polyfoam pile barriers under harmonic excitation of 10 Hz and 15.6 Hz respectively. When the incident SW frequency is 10 Hz, the displacement amplitude does not attenuate significantly along the x-axis and even amplifies between pile barriers, which indicates that the incident wave energy outside the SWBG continues to propagate through pile barriers as in Figure 9b. On the contrary, the incident SW with an excitation frequency

of 15.6 Hz has almost total reflection (see in Figure 9d), which can be observed from the s standing wave pattern formed on the overlap of the incident wave and reflection wave. In addition, a strong exponential decay is observed (see Figure 9c) within the pile array due to the characteristics of Bragg destructive interference. In short, when the frequency zones of the incident waves fall into the SWBG of the ideal periodic structure, the finite arrays of the pile barrier can also effectively prevent the propagation of SWs.

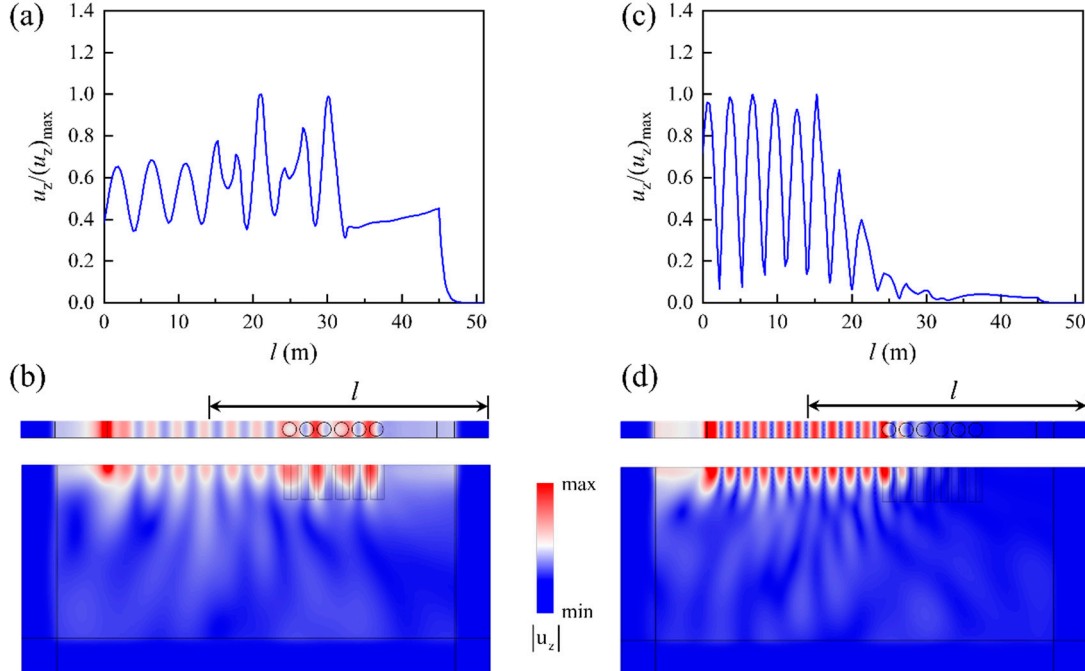

**Figure 9.** The normalized amplitude of the z-displacement at the free surface and wave field distribution of six rows of polyfoam pile barriers under different excitation frequencies. (**a**,**b**) f = 10 Hz; (**c**,**d**) f = 15.6 Hz.

## 4. Parametric Investigation

To evaluate the attenuation effect of the periodic piles, parametric analyses are conducted to study the key parameter which controls the SWBGs of periodic piles. In principle, our works adopt the method of univariate parameter analysis, i.e., only a single variable to be varied while the other parameters remain stable.

Consequently, the influences of pile spacing and pile radius on LBF, UBF, and WBG are performed through the dispersion analysis of unit cell, and the influences of pile number, pile length, and source distance on isolation efficiency are investigated by transmission analysis of a finite array of piles. Detailed analysis results are shown in the subsequent sections.

### 4.1. Influence of Pile Spacing

Taking pile radius $r = 1.2$ m and pile length $h_1 = 2a$ as an example, Figure 10 presents the variation trend of LBF, UBF, and WBG at the boundary point X in the first Brillouin zone with pile spacing. It is obvious that both LBF and UBF of the SWs decrease monotonically with the pile spacing increases, and the first SWBG, namely, WBG of the pile barrier also decreases as the pile spacing increases. Simultaneously, it should be noted that the center frequency of the first band gap of Bragg scattering periodic structure is generally located near $c/2a$, that is, the wavelength corresponding to the center frequency of the first band gap is about two times that of the lattice constant. Increasing pile spacing is equivalent to increasing the periodic constant. The Bragg scattering condition is also confirmed by the phenomenon in Figure 10. Thus, the SWBG at low frequencies can be realized by increasing the periodic constant but the bandwidth is also limited.

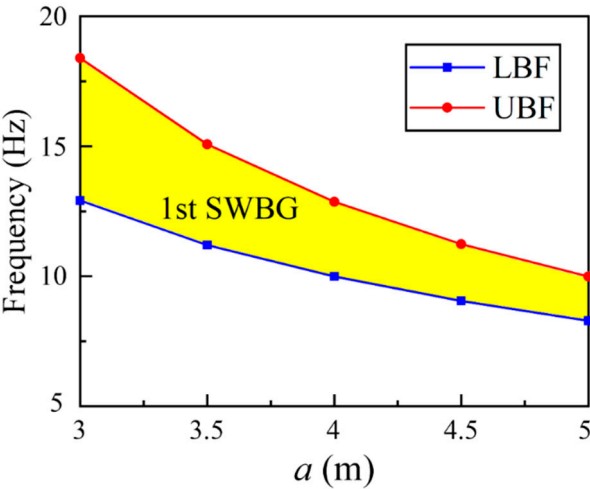

**Figure 10.** Influence of pile spacing. ($r = 1.2$ m, $h_0 = 20a$, $h_1 = 2a$).

### 4.2. Influence of Pile Radius

Keeping $a = 3$ m and $h_1 = 2a$ unchanged, the influence of pile radius on LBF, UBF, and the first SWBG is shown in Figure 11. Parameter β in Figure 11 represents the ratio of pile diameter to periodic constant. It is observed that the UBF and WBG increase monotonically with the pile radius increase, inversely, the LBF of 1st SWBG decreases with the pile radius increase. This means that increasing pile radius is beneficial to vibration reduction. In other words, a larger pile radius could widen the WBG of SWBG and thus take more broadband characteristics of the periodic pile barriers. Without changing the lattice constant, increasing the pile radius corresponds to increasing the volume of the scatterer, which is more conducive to the occurrence of destructive interference.

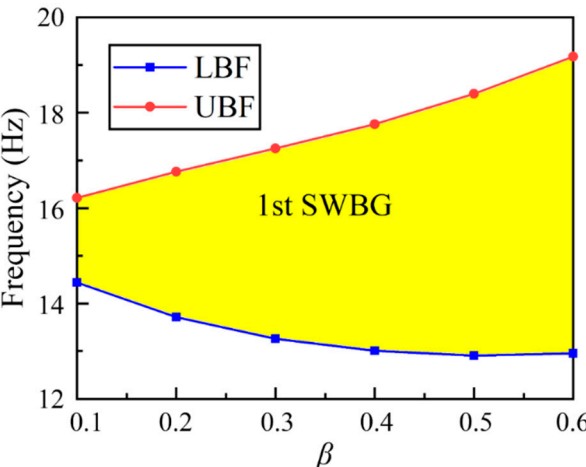

**Figure 11.** Influence of pile radius. ($a = 3$ m, $h_0 = 20a$, $h_1 = 2a$).

### 4.3. Influence of the Number of Piles

The influence of the number of piles on the attention performance for SWs is studied by using a transmission model. The geometrical parameters of the pile barriers are as follows: $a = 3$ m, $r = 1.2$ m, $l_1 = 10a$, $h_0 = 20a$, and $h_1 = 2a$. Figure 12 demonstrates the effect of the number of piles on AAF. Herein, the averaged AAF is suggested for evaluating the mitigation capability in the SWBG, which can be expressed as follows:

$$\overline{\text{AAF}} = \frac{\int_{f_1}^{f_2} \text{AAF} df}{f_1 - f_2} \tag{14}$$

where $f_1$ and $f_2$ are the LBF and UBF of the first SWBG, respectively. Similar to the AAF, the larger the amplitude of $\overline{\text{AAF}}$, the more obvious the isolation effect. As shown in Figure 12a–d, increase in the number of piles triggers a significant increase of $\overline{\text{AAF}}$ when the number of piles varies from 2 to 8, which indicates that the increase of the number for piles is beneficial for improving screening efficiency. This phenomenon can be well explained by the PBC in Equation (9), that is, the displacement vector is reduced by $e^{-\text{Im}(k)a}$ when the SWs propagate one unit cell, where $\text{Im}(k)$ indicates the imaginary part of the corresponding wave vector at a certain frequency inside the SWBG. It can be found that when the number of piles reaches 6, $\overline{\text{AAF}}$ is about $-20$, which means that the displacement amplitude of the output area within the SWBG is 1/10 of the reference model. Considering the isolation performance of SWs and the economic benefit, it is advisable to choose six rows of piles as wave barriers. Additionally, increasing the number of piles has little improvement on the WBG, as shown in the yellow shadow in Figure 12.

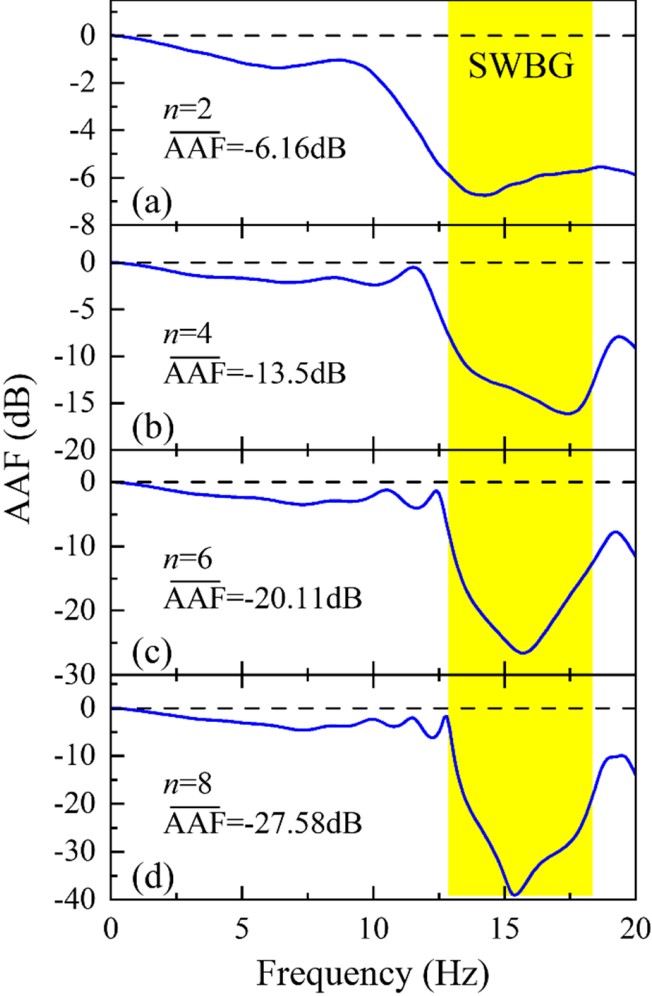

**Figure 12.** Influence of the number of piles. ($a$ = 3 m, $r$ = 1.2 m, $l_1$ = 10a, $h_0$ = 20a, and $h_1$ = 2a).

### 4.4. Influence of Pile Length

Keeping other parameters unchanged, Figure 13 shows the influence of pile length on the AAF. The same averaged AAF is also defined to evaluate the mitigation capability in the SWBG. As depicted in Figure 13, increasing pile length from 0.5a to 2a can significantly improve the attenuation effectiveness. However, with the pile length increased from 2a to 5a, the $\overline{\text{AAF}}$ is stable. In other words, when the pile length reaches the attenuation expectation, i.e., $\overline{\text{AAF}}$ = $-20$ dB, increasing the pile length has little influence on the shielding effect. The reason is that the SW motions mainly concentrate in the thin layer of

soil near the free surface. Therefore, for periodic polyfoam-filled piles, increasing the pile length to a certain extent ($h$ = 2a in this study) could significantly improve the attenuation performance of SWs, but increasing pile length unlimitedly, namely, h > 2a, has no further isolation effect.

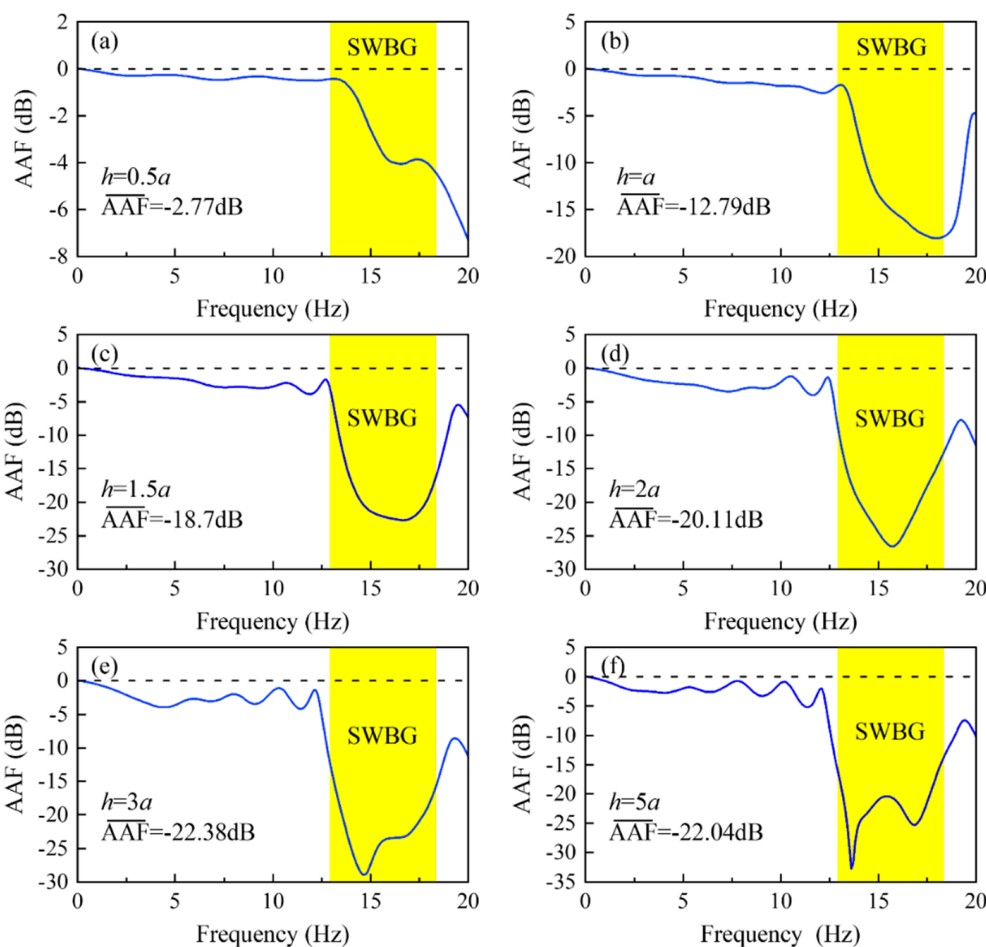

**Figure 13.** Influence of pile length. AAF of six rows of fly ash pile barriers is calculated in frequency domain analysis. ($a$ = 3 m, $r$ = 1.2 m, $l_1$ = 10a, and $h_0$ = 15a).

### 4.5. Influence of Focal Distance

The influence of focal distance $l_1$ of the pile barriers on the AAF is shown in Figure 14. In this section, the source distance $l_1$ is assumed to be varied from 6a to 12a, and the geometrical parameters of the periodic structure are selected as $a$ = 3 m, $r$ = 1.2 m, $h_0$ = 20a, and $h_1$ = 2a. As shown in Figure 14, an increase in source distance cannot improve the average AAF, which indicates that the source distance has little effect on reduction of SWs in SWBG. The reason for the phenomenon is that the localized effects of surface waves, in other words, the elastic strain energies of surface waves, mainly focus on the thin layer. Simultaneously, the conclusions also indicate that the SW intensity is caused by a vertical harmonic load in the far-field basically, which also confirms the attenuation of SWs as the reason for the existence of SWBG.

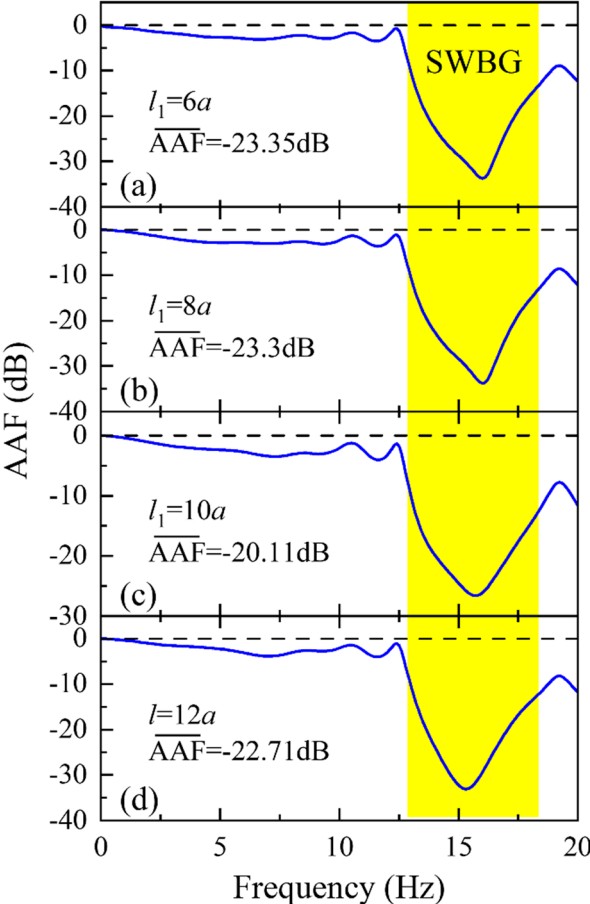

**Figure 14.** Influence of focal distance. AAFs of six rows of fly ash pile barriers are calculated in frequency domain analysis. ($a$ = 3 m, $r$ = 1.2 m, $h_0$ = 15a, and $h_1$ = 2a).

## 5. Transient Analysis

In this section, transient analysis of the periodic pile barriers on isolation effect is conducted by using COMSOL software. The model setup and mesh quality used for transient analysis are similar to that of the transmission analysis model depicted in Figure 7. In addition, Low-reflective-boundary (LRB) conditions are used as the lateral boundaries of the model in place of PML to prevent wave reflections. As aforementioned, the LBF, UBF, and WAZ of the periodic polyfoam pile barriers are 14.68 Hz, 18.54 Hz, and 3.86 Hz, respectively. The artificial wave and seismic wave are used as the vertical excitation load to carry out the analysis. After that, the accelerations at the output area with or without pile barriers are obtained and compared to verify the SW attenuation performance of the periodic barriers.

### 5.1. Harmonic Wave

Harmonic waves with natural frequencies of 10 Hz (outside of SWBG) and 15.6 Hz (inside of SWBG) are applied as incident SW acting on the free surface. $\omega$ is the angular frequency, which can be calculated by equation $\omega = 2\pi f$. Figure 15 shows the vertical acceleration response at the output areas with and without six rows of fly ash pile barriers. From 15a, the dominant frequency of the incident SW is 10 Hz and out of the range of SWBG, which is inconsistent with the AZ frequency range of the periodic fly ash pile barriers. Thus, the normalized vertical acceleration at the output area with the periodic barrier is nearly the same as the results without periodic pile barriers. Conversely, Figure 15b shows the normalized acceleration response with and without periodic pile barriers when the predominant frequency falls into the frequency band gap of the pile barrier. It can be seen

that the results of the pile barriers can significantly attenuate the acceleration response corresponding to the case without pile barriers when the main frequency falls into the AZ. As mentioned above, the periodic pile barriers can significantly reduce the acceleration response when the predominant frequency falls into the frequency band gap.

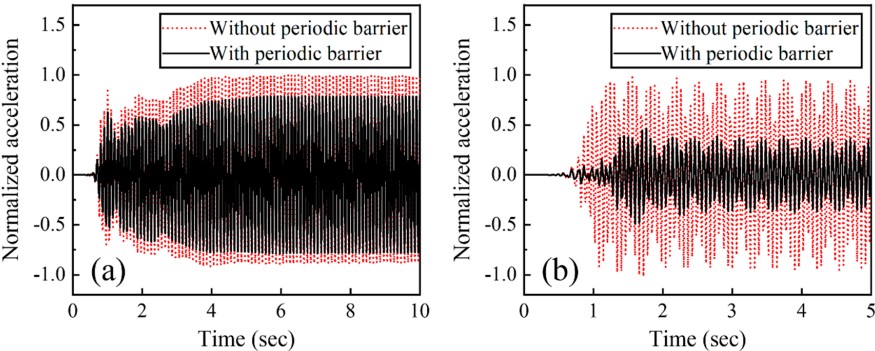

**Figure 15.** The vertical acceleration response at the output areas with and without six rows of fly ash pile barriers. (**a**) f = 10 Hz; (**b**) f = 15.6 Hz. ($a$ = 3 m, $r$ = 1.2 m, $l_1$ = 10a, $h_0$ = 15a, and $h_1$ = 2a).

### 5.2. Artificial Wave

To verify the attenuation performance, an artificial wave is generated by using the normalized combined frequency displacement time history which can be expressed as follows:

$$\begin{cases} u(t) = \frac{1}{u_{\max}} \sum_{i=1}^{n} u_i \sin(2\pi f_i t + \phi_i), \ \ i = 1, 2, \cdots, 30 \\ f_i \in (12.91, 18.39)\text{Hz} \end{cases} \tag{15}$$

where the amplitude of single-frequency displacement $u_i$ represents a random number between 0 and 1, the phase $\phi_i$ is a random number between 0 and $2\pi$, and $u_{\max}$ indicates the maximum value of the combined displacement time history.

Then, the artificial wave produced by Equation (15) is used as a vertical displacement applied at the excitation source, as shown in Figure 16a. As expected, the dominant frequency of this SW is in the range of the SWBG (12.91–18.39 Hz) in the pile barriers. Figure 16b shows the normalized acceleration at the output area with and without six rows of polyfoam pile barriers. From Figure 15b, one can find that the SW at the output area without the periodic barrier is larger than that of the structure with a periodic pile barrier, which obviously indicates that the periodic pile barrier is capable of mitigating the surface waves significantly when the domain frequency of the waves falls into the SWBG. Especially, it can be found that the maximum value of the normalized acceleration is reduced by 60.3% compared to that of the model without periodic barriers.

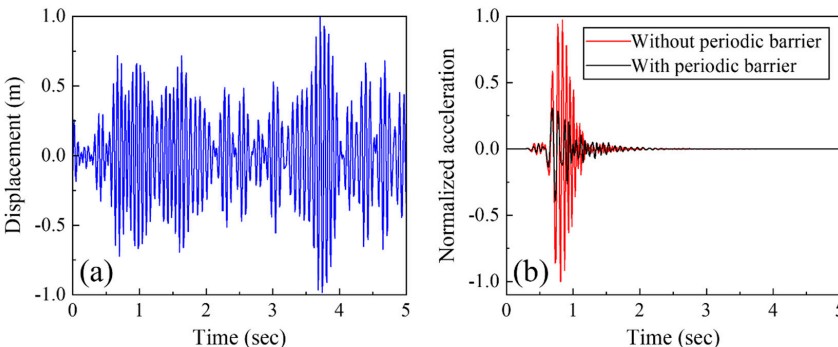

**Figure 16.** Transient analysis results of the artificial wave. (**a**) Displacement time history of the artificial wave; (**b**) vertical acceleration response at the output areas with and without six rows of fly ash pile barriers ($a$ = 3 m, $r$ = 1.2 m, $l_1$ = 10a, $h_0$ = 15a, and $h_1$ = 2a).

### 5.3. Seismic Wave

A seismic wave record obtained from the Whittier Narrows earthquake event is also used to investigate the isolation effect of the finite number of pile barriers. The acceleration history, Fourier spectra, and corresponding displacement history of the seismic record are presented in Figure 17. It can be seen from the Fourier spectra in Figure 17c that the main frequencies of incident signals are distributed at 6.3–8.4 Hz and 11.7–15.4 Hz, among which the second frequency band is located inside the first SWBG.

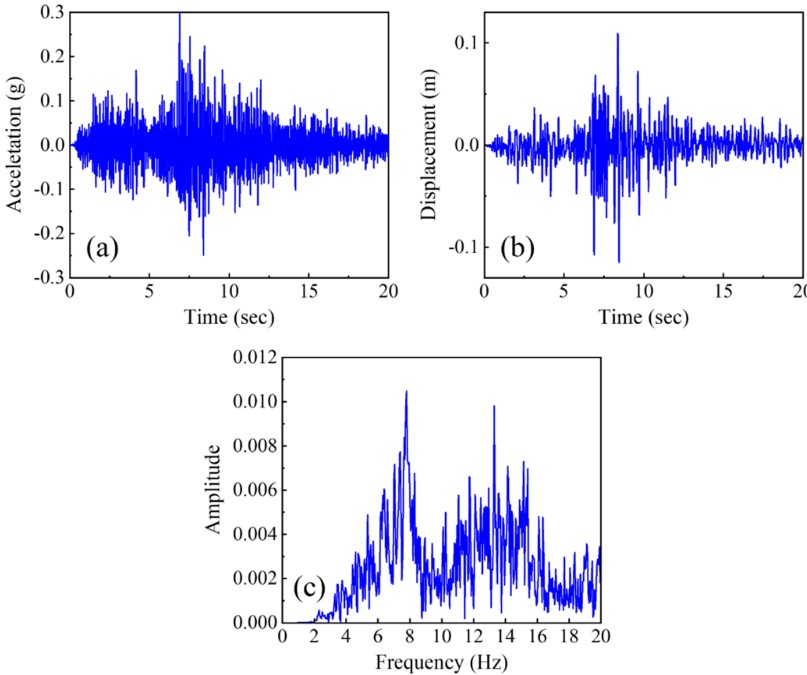

**Figure 17.** Seismic records. (**a**) Acceleration history; (**b**) the corresponding displacement history, and (**c**) Fourier spectra.

Subsequently, Figure 18 shows the transient response of the model with (black lines) and without (red lines) of six rows of pile barriers subjected to seismic waves. The results show that the seismic response is mitigated to some extent with the presence of pile barriers. What is more, the reduction of the peak acceleration with six rows of pile barriers is about 45.4% compared with that without pile barriers. From the Fourier spectra, as shown in Figure 18b, the frequency ranges of seismic reduction agree well with the first SWBG of the periodic pile barriers.

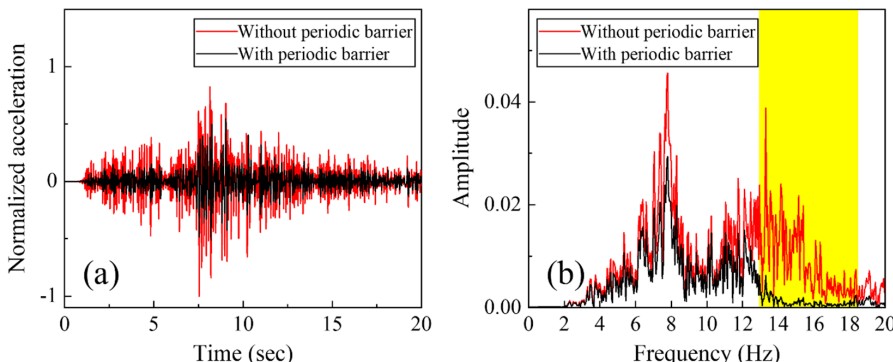

**Figure 18.** Transient response of the model subjected to the seismic waves. (**a**) Vertical acceleration response at the output areas with and without six rows of fly ash pile barriers; (**b**) the corresponding Fourier spectra. ($a = 3$ m, $r = 1.2$ m, $l_1 = 10a$, $h_0 = 15a$, and $h_1 = 2a$).

As aforementioned, the results of surface wave reduction confirm the presence of SWBG in periodic pile barriers and the potential application of SW attenuation in civil engineering.

## 6. Conclusions

In this work, a post-processing technique is introduced to distinguish the Rayleigh modes from all the mixed eigenmodes. A 3-D transmission calculation model is also developed to illustrate that the attenuation zone of a finite number of piles is consistent with the SWBG. Subsequently, the influences of different parameters for pile barriers are parametrically analyzed and discussed. Finally, transient analysis of the periodic pile barriers on isolation effect is performed. Some important conclusions were obtained as follows:

1. For the Bragg scattering periodic structure, such as the buried pile barrier, the energy distribution parameters method can effectively identify all the SW modes while the acoustic cone criterion fails, which may be due to the leakage of some SWs into BWs, resulting in evanescent waves.
2. By comparing the dispersion curves of periodic piles composed of different inclusions, it can be found that lower stiffness and density are more conducive to the generation of low-frequency SWBG. Appropriate design of pile barriers is expected to obtain low-frequency and broadband attenuation.
3. Pile spacing has a strong influence on LBF, UBF, and WBG of the periodic pile barriers. Both LBF and UBF of the first SWBG decrease monotonically with pile spacing increase, narrow pile spacing can widen the WAZ and results in more broadband performance.
4. A certain number of piles results in a significant increase in the periodic pile barriers. However, a further increase in the number of piles cannot widen the WBG and the isolation effect obviously cannot improve.
5. Due to the unique properties of Rayleigh waves, obvious attenuation effect can be observed only when the pile length is increased within a certain range.

**Author Contributions:** Conceptualization, C.Z. (Chunfeng Zhao); Data curation, Y.W.; Investigation, C.Z. (Chunfeng Zhao) and C.Z. (Chao Zeng); Methodology, C.Z. (Chunfeng Zhao); Validation, C.Z. (Chunfeng Zhao) and W.B.; Writing—original draft, C.Z. (Chao Zeng); Check—J.D. All authors have read and agreed to the published version of the manuscript.

**Funding:** This research was funded by the Scientific Research Fund of Institute of Engineering Mechanics, China Earthquake Administration (Grant No. 2020EEEVL0404), and National Natural Science of China (Grant Nos. 52278302, 51508148).

**Data Availability Statement:** The data presented in this study are available on request from the authors.

**Conflicts of Interest:** The authors declare no conflict of interest.

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
