# Peer review of "Theoretical and Numerical Study on the Pile Barrier in Attenuating Seismic Surface Waves"

_buildings, doi:10.3390/buildings12101488_

Round 1

Reviewer 1 Report

The authors have considered the pile spacing, length, count, and material properties as parameters and explored the bounds and range of band gap (the frequencies at which waves would attenuate) using the Flochet-Bloch theory for periodicity.  Further, they have also presented the results from a time domain perspective.   The results although look similar to those of published works, probably the multi-row that authors talk about could be the novelty they refer to.  However, it must be noted here that by default the 2-D theory of dispersion does include multi-row.

Author Response

Reviewer 1

The authors have considered the pile spacing, length, count, and material properties as parameters and explored the bounds and range of band gap (the frequencies at which waves would attenuate) using the Flochet-Bloch theory for periodicity. Further, they have also presented the results from a time domain perspective. The results although look similar to those of published works, probably the multi-row that authors talk about could be the novelty they refer to. However, it must be noted here that by default the 2-D theory of dispersion does include multi-row.

Response: Thanks for reviewer’s comments. We have rewritten the 2-D theory of dispersion that include the multi-row of pile barriers. The multi-row of pile barrier can be assumed as one row arranged periodically and defined a pile unit cell which can be describe by 2D-theory.

Reviewer 2 Report

Respected Authors   I appreciate your work and I believe it deserves to be published in a quality journal. The issue of possible mitigation (attenuation) of vibrations is very vital and attracts attention worldwide.   I have some suggestions concerning the structure of your study and a bunch of editorial comments listed below.   1. Your study looks "very theoretical". I've read it carefully and I couldn't find clear reference to "real world" applications of your studies. I can imagine that your ideas might be applied for reduction of dynamic impact of railways and/or to mitigate harmful impact of geotechnical works (piling, impulse compaction, vibratory roller, horizontal jacking works. Please try to develop your motivation presented in the introductory part referring to works somehow describing possible ways of validation of your findings e.g. DOI: 10.3390/s20071938   2. That also considers imposed ranges of dynamic impact (concerning frequencies and measured accelerations/velocities). ). I am mining engineer but I was once involved in a project devoted to impact of impulse compaction of large deposits on the surrounding area. I know that many authors tried to juxtapose some information concerning this impact e.g. DOI: 10.1051/e3sconf/20199703026   3. You mention numerous technologies to attenuate harmful dynamic impact. Please provide a more accurate description of current “State of the Art” and refer individually to cited papers. Using “cluster citations” like: [1-3], [5-9], [11-14], [17-22] does not allow the Reader to understand the relevance and importance of referenced literature for your study. Please check again DOI: 10.1088/1742-6596/1425/1/012202 . Anyway, please try to introduce the content of your references individually.   4. Please check cautiously the format of your references, as they seem not to be formatted to the MDPI template.   Dear Authors   I pointed some perspective references for your study. I did not co-author them and I have no personal gain from my suggestions. I do not consider my proposal as mandatory, however I believe that widening of your reference list with international “open-access” sources may attract Readers attention (after indexation in databases) and could rise the citing potential of your study.   I marked a major revision just to give you some more time for developing your motivation and maybe for providing some reservations concerning direct applicability of your study. Presented paper, when slightly improved, will certainly be suitable for publication.   Sincerely

Best wishes,

Author Response

Reviewer 2

I appreciate your work and I believe it deserves to be published in a quality journal. The issue of possible mitigation (attenuation) of vibrations is very vital and attracts attention worldwide. I have some suggestions concerning the structure of your study and a bunch of editorial comments listed below.

  1. Your study looks "very theoretical". I've read it carefully and I couldn't find clear reference to "real world" applications of your studies. I can imagine that your ideas might be applied for reduction of dynamic impact of railways and/or to mitigate harmful impact of geotechnical works (piling, impulse compaction, vibratory roller, horizontal jacking works. Please try to develop your motivation presented in the introductory part referring to works somehow describing possible ways of validation of your findings e.g. DOI: 10.3390/s20071938

Response: Thanks for reviewer’s comments, we have revised our introduction and cited the reference that the reviewer provided, which can enrich our findings.

  1. That also considers imposed ranges of dynamic impact (concerning frequencies and measured accelerations/velocities). ). I am mining engineer but I was once involved in a project devoted to impact of impulse compaction of large deposits on the surrounding area. I know that many authors tried to juxtapose some information concerning this impact e.g. DOI: 10.1051/e3sconf/20199703026

Response: Thanks for reviewer’s comments, we have revised our introduction and cited the reference that the reviewer provided, which can enrich our findings.

  1. You mention numerous technologies to attenuate harmful dynamic impact. Please provide a more accurate description of current “State of the Art” and refer individually to cited papers. Using “cluster citations” like: [1-3], [5-9], [11-14], [17-22] does not allow the Reader to understand the relevance and importance of referenced literature for your study. Please check again DOI: 10.1088/1742-6596/1425/1/012202 . Anyway, please try to introduce the content of your references individually.

Response: Thanks for reviewer’s comments, we have revised and simplified the references that cited in cluster citations in introduction. The detailed information is marked in red in the revision.

  1. Please check cautiously the format of your references, as they seem not to be formatted to the MDPI template.

Response: Thanks for reviewer’s comments. We have rewritten the reference according to the MDPI template.

  1. Dear Authors I pointed some perspective references for your study. I did not co-author them and I have no personal gain from my suggestions. I do not consider my proposal as mandatory, however I believe that widening of your reference list with international “open-access” sources may attract Readers attention (after indexation in databases) and could rise the citing potential of your study.

Response: Thanks for reviewer’s insight comments about our research, we have cited the references which can wide our studies in future.

  1. I marked a major revision just to give you some more time for developing your motivation and maybe for providing some reservations concerning direct applicability of your study. Presented paper, when slightly improved, will certainly be suitable for publication.

Response: Thanks for reviewer’s comments, we have revised the paper following the reviewer’s advice. The detailed information is marked in red in the revision.

Round 2

Reviewer 2 Report

Respected Authors

I find presented version of reviewed paper good enough for publication in BUILDINGS. It touches a very actual and vital issue that should guarantee the citing potential of the study. 

I have no further comments.

Sincerely